# A Dynamic Position Embedding-Based Model for Student Classroom Complete Meta-Action Recognition

**DOI:** 10.3390/s24165371

**Published:** 2024-08-20

**Authors:** Zhaoyu Shou, Xiaohu Yuan, Dongxu Li, Jianwen Mo, Huibing Zhang, Jingwei Zhang, Ziyong Wu

**Affiliations:** 1School of Information and Communication, Guilin University of Electronic Technology, Guilin 541004, China; guilinshou@guet.edu.cn (Z.S.); 22022303156@mails.guet.edu.cn (X.Y.); jwmo@guet.edu.cn (J.M.); 2Guangxi Wireless Broadband Communication and Signal Processing Key Laboratory, Guilin University of Electronic Technology, Guilin 541004, China; 3School of Computer and Information Security, Guilin University of Electronic Technology, Guilin 541004, China; zhanghuibing@guet.edu.cn; 4Guangxi Key Laboratory of Trusted Software, Guilin University of Electronic Technology, Guilin 541004, China; gtzjw@hotmail.com (J.Z.); wuziyong@unn.edu.cn (Z.W.)

**Keywords:** meta-action recognition, Video Swin Transformer, dynamic positional embedding, ViT3D

## Abstract

The precise recognition of entire classroom meta-actions is a crucial challenge for the tailored adaptive interpretation of student behavior, given the intricacy of these actions. This paper proposes a Dynamic Position Embedding-based Model for Student Classroom Complete Meta-Action Recognition (DPE-SAR) based on the Video Swin Transformer. The model utilizes a dynamic positional embedding technique to perform conditional positional encoding. Additionally, it incorporates a deep convolutional network to improve the parsing ability of the spatial structure of meta-actions. The full attention mechanism of ViT3D is used to extract the potential spatial features of actions and capture the global spatial–temporal information of meta-actions. The proposed model exhibits exceptional performance compared to baseline models in action recognition as observed in evaluations on public datasets and smart classroom meta-action recognition datasets. The experimental results confirm the superiority of the model in meta-action recognition.

## 1. Introduction

Since student classroom behaviors are composed of a series of meta-actions, the accurate identification of the meta-actions occurring in the classroom is necessary for the individualized adaptive interpretation of student behavior. Traditional action recognition methods utilize end-to-end learning frameworks to automatically extract feature representations that reflect the latent semantic information of video content. ResNet employs residual connections to enhance the extraction of visual information, mitigating problems like overfitting, vanishing gradients, or exploding gradients that may arise while training deep neural network models. The attention mechanism of the Vision Transformer is employed to extract a wide range of essential temporal information from sequences of video frames, allowing the model to acquire more distinct feature representations. However, existing methods neglect the spatial–temporal coherence of actions during feature extraction, limiting recognition accuracy and incurring substantial computational costs when processing high-resolution videos.

To address these challenges, Shou et al. [1] proposed a student behavior detection model based on an improved AIA network, enabling teachers to focus on the learning status of each student. Nevertheless, individual characteristics extracted from the video are inadequate for fully capturing the entirety of the action information. Lin et al. [2] introduced an error correction scheme based on pose estimation and human detection techniques, utilizing the coordinates of skeleton keypoints for classroom action recognition [3]. Khan et al. [4] proposed an attention mechanism based on Bidirectional Long Short-Term Memory (BiLSTM), which employs a Dilated Convolutional Neural Network (DCNN) to selectively focus on effective features in input frames for behavior recognition in videos. Liu et al. [5] adapted the Swin Transformer, originally designed for image recognition, to propose a video recognition architecture based on spatiotemporal locality inductive bias, known as the Video Swin Transformer. While the sliding window attention mechanism reduces computational costs, resulting in the loss of some action information, Chen et al. [6] proposed a Visformer with the same computational complexity that exhibits superior performance compared to both transformer-based and convolutional-based models for ImageNet classification and object detection tasks. Hu et al. [7] proposed a multi-layer pooling and selection fusion 3D convolutional network for video classification. This network combines integrated deep global features with information from shallow local feature extraction networks. Employing three different pooling units—spatiotemporal pyramid pooling, adaptive pooling, and attention pooling—obtains various spatiotemporal feature information, which is ultimately concatenated for classification. To explore multi-scale feature fusion for video classification, Huo et al. [8] proposed a unique visual deformation model based on multi-granular and multi-scale fusion to achieve efficient action recognition.

To address the challenge of the Video Swin Transformer losing some motion information due to its sliding window attention mechanism and the inefficiency of self-attention in encoding low-level features, resulting in low accuracy for classroom meta-action recognition, this paper proposes a Dynamic Position Embedding-based Model for Student Classroom Complete Meta-Action Recognition (DPE-SAR) model. The aim is to accurately recognize the complete classroom meta-actions of students. The main contributions of this study are as follows:A novel network architecture called student action recognition (SAR) is introduced, which combines the Video Swin Transformer and ViT3D. By feeding deep features extracted from the Video Swin Transformer network into ViT3D for comprehensive attention computation, the architecture not only enhances the model’s ability to capture action details but also improved the understanding of the overall structure of the action.The DPE-SAR is introduced, employing a Dynamic Position Embedding method for position encoding in SAR. By incorporating zero-padding in the convolutional process using a deep convolutional approach, each element in the data could better comprehend its absolute position by gradually exploring its neighborhood information, strengthening the model’s understanding of the spatial structure of action videos and improving its local capturing capability for spatial–temporal action features.A student classroom meta-action dataset, GUET10, is constructed for smart classroom scenarios. The effectiveness and reliability of the SAR and DPE-SAR models are validated on diverse video data across various action recognition scenarios, providing a new technical approach for smart education and behavior analysis domains.

The remainder of the paper is organized as follows: Section 2 provides a brief review of the related work. Section 3 introduces the materials and methods used for experiments and gives a detailed description of the Dynamic Position Embedding-based Model for Student Classroom Complete Meta-Action Recognition (DPE-SAR). Section 4 shows the experimental results. Section 5 offers an in-depth analysis of the experimental results. Finally, Section 6 concludes the work with an outlook.

## 2. Related Work

Traditional student classroom action recognition methods typically use 2D CNNs applied to image streams. However, these methods struggle to capture the long sequence of temporal dependencies of actions and lack global features, resulting in low student action recognition accuracy. In recent years, many researchers have focused on developing video action recognition models based on 3D CNNs and Vision Transformers; 3D CNNs are widely used in computer vision tasks such as video processing and action recognition [9], capturing spatial and temporal information by applying convolution operations across three dimensions (width, height, and time) of the data [10]. However, 3D CNNs involve a large number of parameters, requiring significant computational resources and longer training times. On smaller datasets, the vast number of parameters can lead to overfitting. In addition, due to their limited receptive fields, 3D CNNs have difficulty capturing long-range dependencies [11].

The Vision Transformer (ViT), a model based on the Transformer architecture, was introduced to computer vision after achieving success in natural language processing (NLP) [12,13]. ViT splits images into patches, linearly embedding them into a sequence, and processes them using the standard Transformer architecture to capture relationships between any two regions in the image [14,15,16], which is typically challenging for convolutional neural networks [17]. Although ViT can capture global dependencies [18,19], the local feature-capturing ability is not as strong as that of convolutional neural networks [20,21].

Furthermore, the self-attention mechanism is inefficient in encoding low-level features, limiting the potential for recognition accuracy [22,23]. The positional encoding in Vision Transformer (ViT) architectures typically encompasses three distinct categories, absolute position encoding [24,25], relative position encoding [26,27], and conditional position encoding, providing valuable supervision for dependencies between different positional elements in the sequence.

Among these, conditional position encoding (CPE) is dynamically generated and conditioned on the local neighborhood of the input tokens, varying with the input size and maintaining the required translation invariance. They enhance adaptability to input shapes and can easily generalize to input sequences longer than those seen during training [28].

In summary, 3D CNN-based backbone networks have a distinct advantage in local feature extraction but exhibit low long-term dependency capabilities. On the other hand, backbone networks based on Transformers have the ability to capture global data but are insufficient for local feature extraction. Absolute and relative position encodings struggle to efficiently encode position information for different input shapes, whereas conditional position encoding effectively addresses this issue. Therefore, this paper extends conditional position encoding [29] and integrates the Video Swin Transformer with ViT3D to propose a new student classroom meta-action recognition model, DPE-SAR. The model uses the Video Swin Transformer as the backbone network, employs Dynamic Position Embedding for conditional position encoding to enhance local feature capturing ability and spatial structure ability of meta-actions, and utilizes ViT3D for full attention computation to associate potential spatial–temporal features during action occurrence. Experiments are conducted on student classroom meta-action recognition in smart classroom scenarios to validate the effectiveness and reliability of the model in accurately recognizing student classroom meta-actions.

## 3. Materials and Methods

### 3.1. Dataset

In this work, public action recognition datasets UCF101 and HMDB51 are used for experiments. The UCF101 dataset is an extension of UCF50 and consists of 13,320 video clips divided into 101 categories. These 101 categories can be divided into five types: body movements, human–human interactions, human–object interactions, playing musical instruments, and sports. The total length of the video clips exceeds 27 h. All video frames are fixed at 25 FPS, with a resolution of 320 × 240. The HMDB51 dataset comprises videos collected from various sources, such as movies and online videos. The dataset consists of 6766 video clips across 51 action categories (e.g., “jump”, “kiss”, and “laugh”), with at least 101 clips in each category.

Since student classroom behaviors are composed of meta-actions, building a student classroom meta-action recognition dataset is essential for providing individualized interpretations of these behaviors. Consequently, this paper constructs the GUET10 dataset in a smart classroom setting. The dataset consists of video clips of student classroom meta-actions recorded in 2023 during a particular course. It includes 1863 video clips across 10 action categories: “read”, “take notes”, “talk”, “tidy hair”, “resting head on hand”, “lie on the desk”, “listen”, “raise a hand”, “stand”, and “use the phone”, with at least 100 clips per category. The frames are fixed at 30 FPS, with a resolution of 640 × 480. The details are shown in Table 1.

### 3.2. Dynamic Position Embedding-Based Model for Student Classroom Complete Meta-Action Recognition (DPE-SAR)

This section provides an overview of the framework of the DPE-SAR model. The input shape of the model is [T, H, W, C]. First, the input undergoes Dynamic Position Embedding (DPE) to obtain the embedded tokens. These tokens are then fed into the backbone network, which consists of four stages with depths of (2, 2, 6, 12). The first three stages are based on the Video Swin Transformer modules, while the fourth stage utilizes the ViT3D module. Finally, the action recognition results are obtained. The overall architecture of the model is illustrated in Figure 1.

Positional information is a crucial cue for describing visual representations. Conditional position encoding (CPE) utilizes convolution operators to implicitly encode positional information, enabling Transformers to handle arbitrary input sizes and enhancing recognition performance. Due to the plug-and-play nature of CPE, it is used as the implementation method for Dynamic Position Embedding (DPE) in this model. The formula is as follows:(1)$$\mathrm{DPE}\left(X\right)=\mathrm{DWConv}\left(X\right)$$

In Equation (Equation 1), DWConv is a type of depthwise convolution method that uses zero padding to enhance adaptability to input shapes. Additionally, by introducing zero padding during the convolution process, DWConv allows each element in the data to gradually explore its neighboring information, thereby improving the understanding of absolute positions within the data. This enhances the model’s ability to comprehend spatial structures.

The foundational Video Swin Transformer module serves as a versatile backbone network for action recognition, demonstrating robust performance across various granularity recognition tasks such as region-level object detection, pixel-level semantic segmentation, and pixel-level image classification. Integrated within the Transformer encoder, the Video Swin Transformer introduces windowed attention to achieve hierarchical structure, locality, and translational invariance. This approach combines the powerful modeling capabilities of the fundamental transformer unit while reducing computational complexity. The network architecture is illustrated in Figure 2.

In Figure 2, the basic block undergoes 3D W-MSA (window-based multi-head self-attention) computation first, followed by layer normalization (LN), residual connection, and then feature selection through MLP layers and normalization. The window-based self-attention computation module lacks inter-window connections, which limits its modeling capability. The sliding window attention calculation employs a shifting window partition method, alternating between two partition configurations in consecutive Video Swin Transformer modules:(2)$${\hat{z}}^{l}=LN\left(\text{3DW-MSA}\left(z^{l-1}\right)\right)+z^{l-1}$$
(3)$$z^{l}=LN\left(MLP\left({\hat{z}}^{l}\right)\right)$$
(4)$${\hat{z}}^{l+1}=LN\left(\text{3DSW-MSA}\left(z^{l}\right)\right)+z^{l}$$
(5)$$z^{l+1}=LN\left(MLP\left({\hat{z}}^{l+1}\right)\right)+{\hat{z}}^{l+1}$$

In Equations (2)–(5), ${\hat{z}}^{l}$ and ${\hat{z}}^{l+1}$ represent the features from (sliding window) window-based self-attention computation and MLP layer output, respectively. The use of the sliding window attention calculation enables the association of features between consecutive frames, which significantly enhances performance in video recognition tasks.

Due to the simultaneous sliding window processing of feature maps in both the temporal and spatial dimensions by the Video Swin Transformer module, some motion information is lost. In contrast, the ViT3D module expands the parameters of ViT, concatenating videos or multiple images into a group input. This allows the model to perform full attention calculation across all small patches in the deep feature maps, effectively connecting the global characteristics of spatiotemporal features of actions. The model structure is illustrated in Figure 3.

After the output feature maps from Stage 3 in Figure 1, the ViT3D module processes them through a Transformer Encoder for full attention computation, extracting attention for each pixel block. Finally, classification is performed via the MLP HEAD layer to obtain the recognition of classroom meta-actions.

## 4. Experimental Section

### 4.1. Experimental Details

The experimental setup of this paper is shown in Table 2.

This paper conducts experiments using the Video Swin Transformer model framework. The AdamW optimizer is used for gradient descent optimization. The learning rate for the backbone network is set to 3 × 10^−5^, with an input frame rate of 8 frames and a shape size reset to 224×224. The batch size is set to 64, and the total number of iterations is 500 epochs.

### 4.2. Experimental Evaluation of the SAR Action Recognition Model

To verify the effectiveness and advantages of the SAR architecture combining the Video Swin Transformer and ViT3D proposed in this paper, experiments are conducted using the Video Swin Transformer (Swin) as the baseline model. These experiments are performed on two public datasets, UCF101 and HMDB51, as well as a smart classroom scenario dataset, GUET10, which consists of student classroom meta-action data. The evaluation metrics include Top1 accuracy (Top1 Acc) and mean class accuracy (mean class acc). The “mean class accuracy” is the average accuracy across all classes, calculated by summing the accuracy of each individual class and then dividing by the total number of classes. Comparative analysis with the baseline model is illustrated in Figure 4, Figure 5 and Figure 6.

From Figure 4, Figure 5 and Figure 6, each point on the x-axis represents 10 epochs, with accuracy plotted on the y-axis. The graphical representation provides a visual comparison of the performance curves of two models on respective datasets. Specifically, Figure 4 and Figure 5 present the results trained on the public datasets UCF101 and HMDB51. Comparative analysis indicates that the SAR model outperforms the baseline model with improvements of 0.1089% and 3.934% in Top1 Acc, and 1.133% and 4.915% in mean class accuracy, respectively. Figure 6 shows the results trained on the educational scenario dataset GUET10, showing improvements of 4.986% in Top1 Acc and 5.926% in mean class accuracy compared to baseline performance.

### 4.3. Experimental Evaluation of DPE-SAR Action Recognition Model

To further explore and validate the effectiveness of the DPE-SAR model, experiments were conducted using the same model and methodology on the public datasets UCF101, HMDB51, and GUET10. The performance curves are illustrated in Figure 7, Figure 8 and Figure 9.

Figure 7, Figure 8 and Figure 9 show performance curves, where each point on the x-axis represents 10 epochs, with accuracy plotted on the y-axis. Specifically, Figure 7 and Figure 8 show training results on the public datasets UCF101 and HMDB51. Comparative analysis reveals that the action recognition model of DPE-SAR outperforms the baseline model with improvements of 1.54% and 4.011% in Top1 Acc, and 1.599% and 5.274% in mean class accuracy, respectively. Figure 9 illustrates the training results on the educational scenario dataset GUET10, showing that the DPE-SAR model achieves improvements of 6.925% in Top1 Acc and 7.072% in mean class accuracy compared to the baseline performance.

## 5. Discussion

Compared to the baseline model, experiments were conducted to verify the effectiveness of the SAR model and DPE-SAR model in video action recognition on two public datasets, UCF101 and HMDB51. In addition, to verify the effectiveness of the proposed models, experiments were conducted on the educational scenario dataset GUET10. The evaluation is based on the Top1 accuracy (Top1 Acc) and mean class accuracy (mean class acc). The experimental results are presented in Table 3.

The performance metrics of the proposed SAR and DPE-SAR models on the datasets UCF101, HMDB51, and GUET10 compared to the baseline model Video Swin Transformer are presented in Table 3. The analyses are as follows:Experiments were conducted on the outdoor sports datasets UCF101 and HMDB51, as well as the educational scenario dataset GUET10, to evaluate the proposed models. Both the SAR and DPE-SAR models achieved higher Top1 Acc and mean class acc compared to the baseline model. These results effectively validate the superiority of the proposed models in action recognition tasks.After comparing and analyzing the SAR and Video Swin Transformer models, it is evident that the SAR model significantly improves both key performance metrics, Top1 Acc and mean class acc. The result indicates that the SAR model achieves superior performance in action recognition tasks due to full attention computation at a deep network stage (Stage 4), enabling it to focus more on the global characteristics of actions. The advantage of this full attention mechanism lies in its ability to comprehensively consider the contextual information of each element in the sequence, rather than being restricted to information within a local window. This enhances the model’s understanding and recognition capability of the overall action structure.By comparing the DPE-SAR and SAR models, we observe further improvements in the Top1 Acc and mean class acc values. The result validates the effectiveness of Dynamic Positional Encoding (DPE) in enhancing the model’s understanding of the spatial structures in action videos. Through the incorporation of Dynamic Positional Encoding, the DPE-SAR model can capture the spatial–temporal features of actions more precisely and better comprehend the temporal–spatial evolution of actions, thereby achieving more precise and accurate predictions in action recognition tasks.

In this paper, the proposed SAR, DPE-SAR, and baseline model Video Swin Transformer are validated on the GUET10 test set. The confusion matrices are illustrated in Figure 10, Figure 11 and Figure 12, where the horizontal axis represents the predicted labels by the models, the vertical axis represents true labels, and the numerical values represent the number of videos.

In Figure 10, Figure 11 and Figure 12, the analysis reveals that compared to the baseline model Video Swin Transformer, both SAR and DPE-SAR achieve higher overall recognition accuracy. All three models tend to misclassify “take notes” as “read book”, which is attributable to the temporal overlap between video clips of these two actions. Although DPE-SAR shows slightly lower accuracy than SAR in recognizing actions such as “lie on the table”, “raise a hand”, and “use the phone”, it achieves the highest overall recognition accuracy, validating its superiority.

To validate the model’s performance, a comparative experiment measured its runtime. The experimental results are presented in Table 4.

Table 4 shows that DPE-SAR demonstrates faster inference times than the baseline model for videos in the 8–32 fps range. However, as the frame rate increases, the inference time of DPE-SAR surpasses that of the baseline model. This discrepancy arises because the computation of all attentions at once in DPE-SAR (Stage 4) requires more time than computing attentions using sliding windows.

**Limitations:** The proposed model primarily focuses on recognizing single-person single-action scenarios, and it has limitations when applied to videos with multiple people and multiple actions. Additionally, the model’s effectiveness has only been validated for low-resolution student action recognition in classroom settings. A more careful analysis is left for future work.

## 6. Conclusions

This paper proposes the Dynamic Position Embedding-based Model for Student Classroom Complete Meta-Action Recognition (DPE-SAR) model to effectively integrate global temporal features of actions by performing full attention computation on the deep features (Stage 4) from the Video Swin Transformer network. In addition, leveraging Dynamic Positional Encoding enhances the model’s understanding of spatial structures in action videos and improves its ability to capture the local spatiotemporal features of actions. The experimental results demonstrate that the proposed model achieves higher recognition accuracy than the baseline model on the public datasets UCF101, HMDB51 and the educational scenario dataset GUET10, verifying the effectiveness of the DPE-SAR model in recognizing meta-actions in classroom settings and its adaptability to smart classroom environments. The proposed model demonstrates its potential for practical applications, providing a valuable reference for future research and applications in comprehensive meta-actions recognition in classrooms and individualized behavioral interpretation for students.

For future work, the developed DPE-SAR model will be further enhanced to handle more complex classroom scenarios, such as multi-student and multi-action recognition.

## Figures and Tables

**Figure 1 sensors-24-05371-f001:**
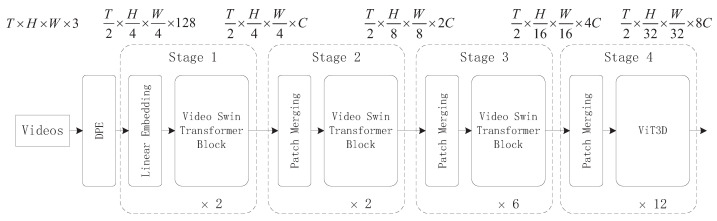
Overall architecture of the DPE-SAR model.

**Figure 2 sensors-24-05371-f002:**
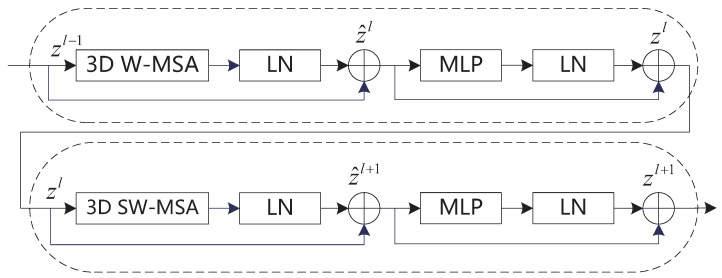
Network diagram of the Video Swin Transformer basic block.

**Figure 3 sensors-24-05371-f003:**
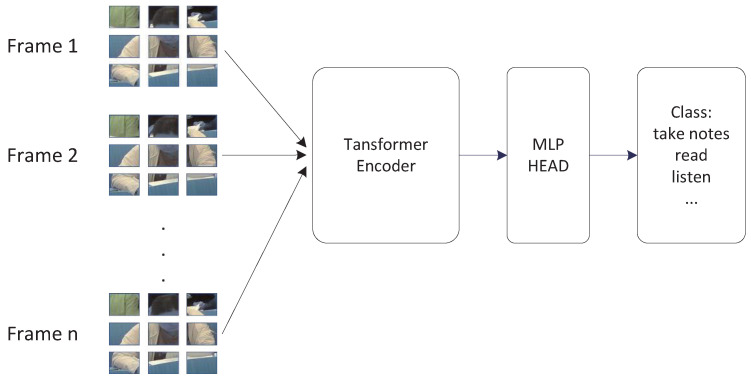
ViT3D structure diagram.

**Figure 4 sensors-24-05371-f004:**
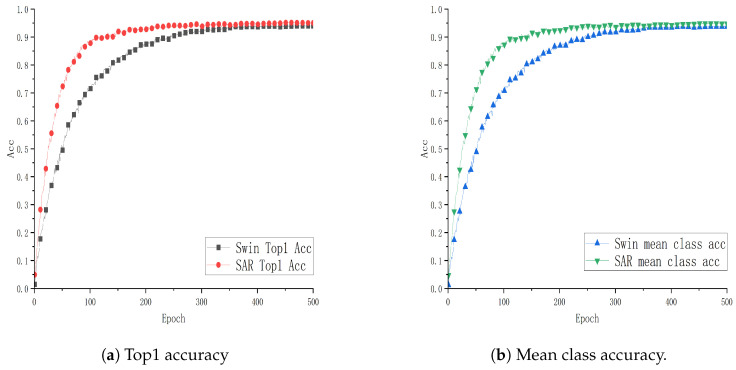
Performance curve of SAR on the UCF101 dataset.

**Figure 5 sensors-24-05371-f005:**
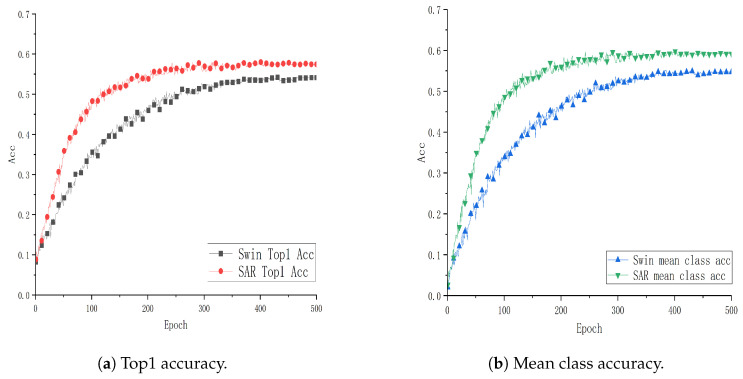
Performance curve of SAR on the HMDB51 dataset.

**Figure 6 sensors-24-05371-f006:**
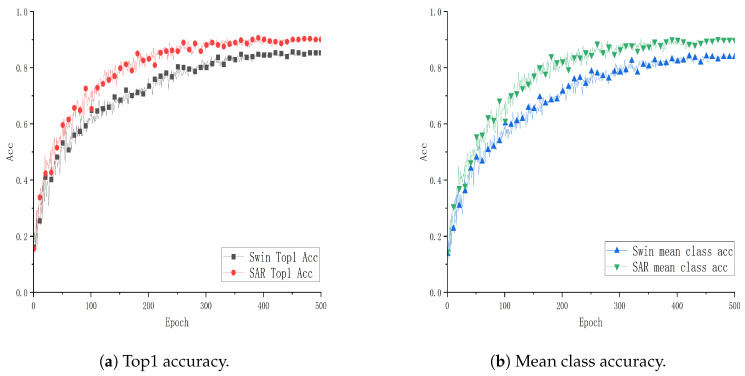
Performance curve of SAR on the GUET10 dataset.

**Figure 7 sensors-24-05371-f007:**
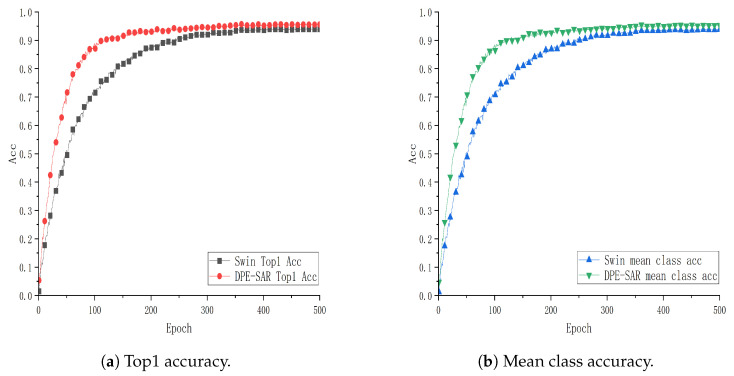
Performance curve of DPE-SAR on the UCF101 dataset.

**Figure 8 sensors-24-05371-f008:**
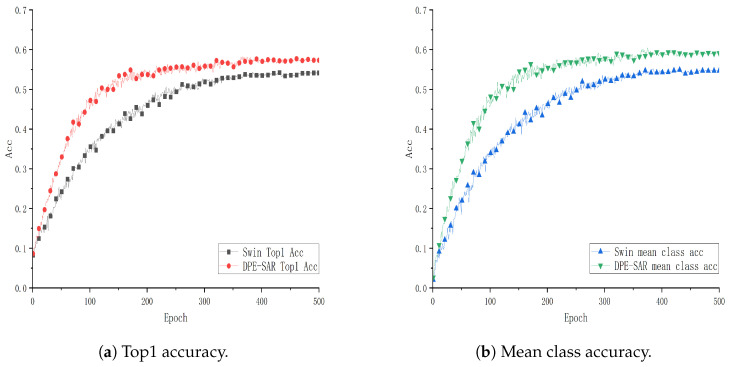
Performance curve of DPE-SAR on the HMDB51 dataset.

**Figure 9 sensors-24-05371-f009:**
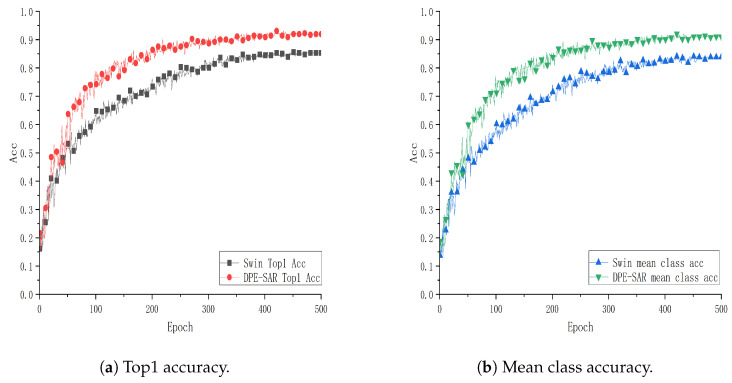
Performance curve of DPE-SAR on the GUET10 dataset.

**Figure 10 sensors-24-05371-f010:**
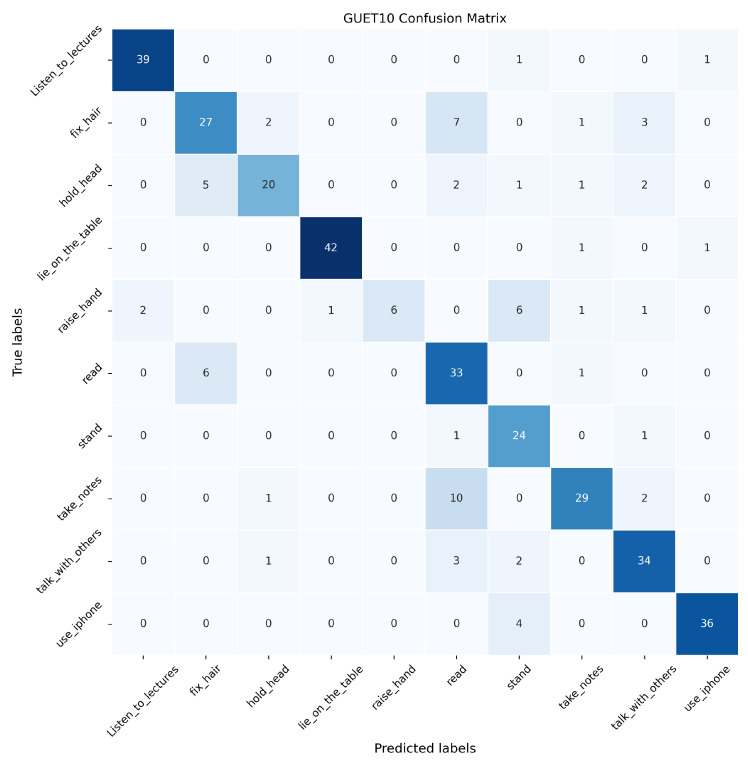
Confusion matrix of Video Swin Transformer on the GUET10 test set.

**Figure 11 sensors-24-05371-f011:**
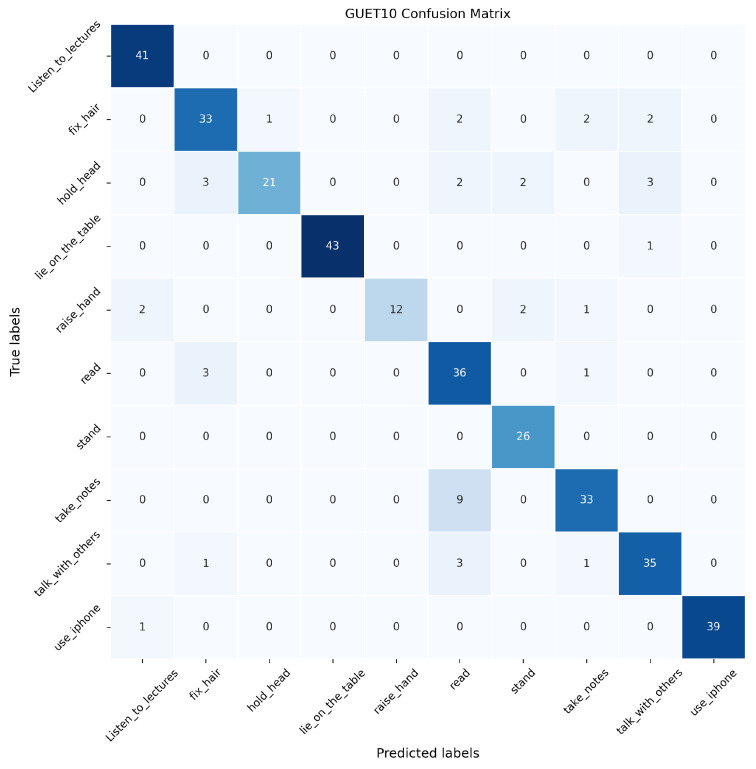
Confusion matrix of SAR on the GUET10 test set.

**Figure 12 sensors-24-05371-f012:**
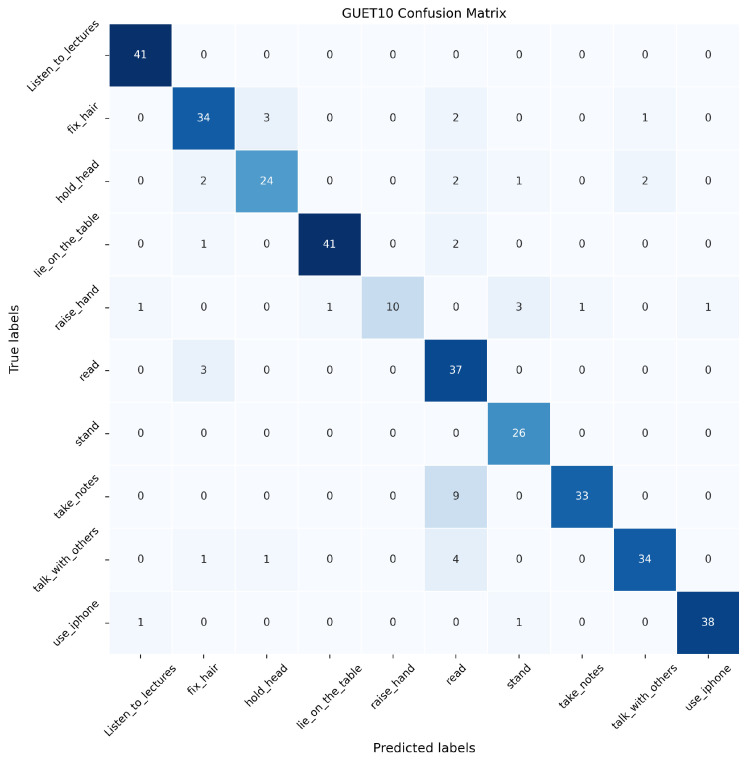
Confusion matrix of DPE-SAR on the GUET10 test set.

**Table 1 sensors-24-05371-t001:** Action recognition dataset.

Dataset	Number of Class	Number of Video Clips	Average Duration per Clip
UCF101	101	100 to 200	3 to 10 s
HMDB51	51	100 to 110	2 to 20 s
GUET10	10	100 to 200	3 to 18 s

**Table 2 sensors-24-05371-t002:** Experimental environment.

Experimental Environment	Environment Configuration
Operating systems	Win11
CPU	Ryzen 9 7950X
Video Cards	GeForce RTX 4090
RAM	64GB
ROM	2T SSD
Programming Languages	Python 3.9
Framework	Pytorch

**Table 3 sensors-24-05371-t003:** Experimental results comparison.

Dataset	Model	Swin	SAR	DPE-SAR
UCF101	Top1 Acc	0.94140	0.95229	**0.95680**
	mean class acc	0.93876	0.95009	**0.95475**
HMDB51	Top1 Acc	0.54269	0.58203	**0.58280**
	mean class acc	0.55156	0.60071	**0.60430**
GUET10	Top1 Acc	0.86150	0.91136	**0.93075**
	mean class acc	0.84993	0.90919	**0.92065**

**Table 4 sensors-24-05371-t004:** With the resolution reset to 224 × 224, the inference time (in seconds) of the two models at different fps on the GUET10 dataset.

Fps	Swin	DPE-SAR
8	0.1436	**0.1406**
16	0.1688	**0.1430**
32	0.2546	**0.2273**
64	**0.3326**	0.3368
128	**0.5667**	0.5916

## Data Availability

The data are available from the corresponding author on reasonable request.

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
