# Peer review of "A Dynamic Position Embedding-Based Model for Student Classroom Complete Meta-Action Recognition"

_sensors, 2024, doi:10.3390/s24165371_

Round 1

Reviewer 1 Report

Comments and Suggestions for Authors

Comments:
The submitted manuscript proposes A DynamicPosition Embedding-based Model for Student Classroom Complete Meta-action Recognition. As a teaching aid module, this model aims to accurately recognize the meta-actions of students in the classroom. The paper exhibits a high overall quality, featuring a novel and unique methodology.

Compared to state-of-the-art approaches, the authors test the proposed Dynamic Position Embedding-based Model for Student Classroom Complete Meta-action Recognition on two public datasets and demonstrate its advantages over current student classroom meta-action recognition models. The technical implementation is sound and explained in detail, which is favorable for reproducibility.

Experiments show that the algorithm in this paper, addressing the limitations of the Video Swin Transformer, integrates dynamic positional embedding and the full attention mechanism of ViT3D, effectively enhancing the recognition accuracy of classroom meta-actions, and providing technical support for the interpretation of students' individual adaptive behaviors. However, in its current form, the manuscript contains some shortcomings. Slight modifications should be made on the following points to argue for a recommendation for publication.

1. There are typos and grammatical errors here as well. It needs to be re-proofread.

2. The working part is solid, but the meaning of each subfigure should be accurately described in the captions to highlight the comparative nature of the experiments.

3. The proposed methodology is highly applicable and I would have liked the concluding chapter to shed light on potential applications and future research in a broader sense.

Comments on the Quality of English Language

English expression needs further improvement.

Reviewer 2 Report

Comments and Suggestions for Authors

1.         The summary of previous researches in the “Introduction” and “related work” are very well in balance between clearly simplified and key information with detailed, so that it is worth to encourage the author who correspond within this part.

2.         In the “Materials and Methods”, the dataset which operated in the lower video resolution and lower frame per second (FPS) condition; however, how to prove the lower video setup can still meaningfully analysis the improved software model for “Student Classroom Complete Meta-action Recognition” application under without hardware error?

3.         If the authors can prove or present a reasonable viewpoint for analyzing the improved model in lower resolution, the total runtime for “Meta-action Recognition Application” should be competitive advantage. Please reveal the runtime of the analyzed procedure. It will be a key information for the real application in the future.

4.         If the authors also think the higher video resolution and higher FPS are necessary into “Meta-Action Recognition” application, and the experimental environment (ex. GPU RTX 4090…etc.) in this article maybe can enough achieve higher performance analysis. Please try to use higher resolution (ex. > 1080p) and higher FPS (ex. > 90Hz) to revised the results in this articles.

5.    As we known, the precision and accuracy is very different definition to evaluate the video recognition. In the row 212 ~ 213, the “Top1 Accuracy (Top1 Acc) and Mean Class Accuracy (mean class acc)” which are represented “Accuracy” and “Precision” in respectively? If not, please define more clearly the metric method. 

Reviewer 3 Report

Comments and Suggestions for Authors

The authors present an approach to student behavior recognition. The neural approach is based on the use of the dynamic positional embedding technique. The results showed greater scores on two public action recognition datasets. The paper is well-structed, the results are clearly presented. The authors are suggested to add an error analysis section for better understanding of the limitations of the proposed approach.

The authors present an approach to student behavior recognition. The neural approach is based on the use of the dynamic positional embedding technique.The authors also present a new neural architecture combining VST and  ViT3D. The experiments are conducted on a dataset of student classroom meta-actions.

The idea of combining VST and ViT3D is intersting. The main motivation of this idea is extracting deep features from the VST network and the use of these features in ViT3D which respresents the three-dimentional transformer-based architecture. The research can be interting for a scientific community since it uses and expands state-of-the-art methods.

The authors state that their approach achieves higher recognition accuracy than the baseline model on three datasets. The paper also presents a new dataset GUET10 consisted of student classroom meta-action videos.

The authors are suggested to add an error analysis section for better understanding of the limitations of the proposed approach. In the currect version of the paper the limitations of the study are not discussed.

In general, the conclusions correspond to the content of the paper. Howerver, the conclusions will be more strong after the providing strong analysis of the model's errors.

The references are appropriate.

The figures and tables are of sufficient quality.

Round 2

Reviewer 2 Report

Comments and Suggestions for Authors

The revised manuscript can be accepted.